# Doubly Dynamic Hydrogel Formed by Combining Boronate Ester and Acylhydrazone Bonds

**DOI:** 10.3390/polym12020487

**Published:** 2020-02-21

**Authors:** Yusheng Liu, Yigang Liu, Qiuxia Wang, Yugui Han, Hao Chen, Yebang Tan

**Affiliations:** 1School of Chemistry and Chemical Engineering, Key Laboratory of Special Functional Aggregated Materials, Ministry Education, Shandong University, Jinan 250100, China; liuyusheng@mail.sdu.edu.cn; 2CNOOC, Ltd., Tianjin Branch, Bohai Oilfield Research Institute, Tanggu Tianjin 300452, China; liuyg@cnooc.com.cn (Y.L.); wangqx17@cnooc.com.cn (Q.W.); hanyg4@cnooc.com.cn (Y.H.)

**Keywords:** hydrogel, dynamic covalent bond, diblock copolymers, self-healing

## Abstract

The incorporation of double dynamic bonds into hydrogels provides an effective strategy to engineer their performance on demand. Herein, novel hydrogels were PREPARED by combining two kinetically distinct dynamic covalent bonds, boronate ester and acylhydrazone bonds, and the synergistic properties of the hydrogels were studied comprehensively. The functional diblock copolymers P(*N*-isopropyl acrylamide-co-*N*-acryloyl-3-aminophenylboronic acid)-b-(*N*-isopropyl acrylamide-co-diacetone acrylamide) (PAD) were prepared via reversible addition−fragmentation chain transfer (RAFT) polymerization. The hydrogel was constructed by exploiting dynamic reaction of phenyboronic acid moieties with polyvinyl alcohol (PVA) and ketone moieties with adipic dihydrazide (ADH) without any catalyst. The active boronate ester linkage endows the hydrogel with fast gelation kinetics and self-healing ability, and the stable acylhydrazone linkage can enhance the mechanical property of the hydrogel. The difference in kinetics endows that the contribution of each linkage to mechanical strength of the hydrogel can be accurately estimated. Moreover, the mechanical property of the hydrogel can be readily engineered by changing the composition and solid content, as well as by controlling the formation or dissociation of the dynamic linkages. Thus, we provide a promising strategy to design and prepare multi-responsive hydrogels with tunable properties.

## 1. Introduction

Hydrogels based on dynamic covalent bonds (DCBs) have attracted significant attention due to their excellent performance in self-healing and multi-responsive materials [1,2,3]. The DCBs combine the robustness of covalent bonding and reversibility of noncovalent bonding in a single system. The stability of the DCB endows the hydrogels with proper mechanical properties, while the dynamic nature of the DCB provides both self-healing and controllable stimuli-responsive properties [4,5]. Recently, the DCBs-based materials with controllable properties and multiple responsiveness have become increasingly popular because of their application prospects in some complex environments [6,7]. To this end, considerable research has been devoted to the incorporation of multiple DCBs in a single system to enhance the performance and complexities of the material [8,9].

Multiple dynamic covalent bonds in materials often complement each other [10]. Compared with the single DCB-crosslinked hydrogels, the integration of dual or multiple DCBs in one system can effectively overcome the limitations of single DCB-crosslinked materials and enhance the performance of hydrogel. The synthesis of a doubly dynamic crosslinking agent based on orthogonal reactions is an efficient strategy for achieving this goal [8]. Multiple dynamic covalent bonds in materials often complement each other [10]. In this case, two or more dynamic chemistries can respond to stimuli independently that enables the functionalities of the materials improved by alternately controlling the different dynamic interaction. In recent study, Zhang and coworkers synthesized a novel type of crosslinker bis(phenylboronic acid carbamoyl) cystamine (BPBAC) and used for the preparation of multifunctional hydrogel [11]. The pH, glucose, and redox triresponsiveness and self-healing ability of the hydrogel were achieved with boronate ester and disulfide exchange reaction. Another promising strategy for endowing materials with multi-dynamic system is to synthesize or modify polymers with various reactive moieties involved in the formation of DCBs. For instance, Chen and coworkers integrated the Diels–Alder click reaction (DA) reaction and acylhydrazone bond in a doubly cross-linked hydrogel used in cartilage tissue engineering by modifying the hyaluronic cidasodium with adipic dihydrazide and furylamine [12]. The DA reaction maintains the hydrogel with good structural integrity and the acylhydrazone bond enables self-healing ability of the hydrogel under physiological conditions. Chen and coworkers incorporated the acylhydrazone and imine bond in a biocompatible hydrogel based on oxidized sodium alginate (OSA) [13]. The excellent self-healing ability and ideal mechanical strength of the hydrogel were obtained from the integration of dual dynamic crosslink system. Recent research also showed that the combination of kinetically distinct dynamic systems could engineer the mechanical behavior of the hydrogels [14]. Combining multiple dynamic bonds into hydrogels has become an important strategy to engineer their performance on demand.

As a representative DCB, acylhydrazone bond, which is formed from acylhydrazine and aldehyde/ketone, have been successfully used to prepare self-healing and stimuli-responsive hydrogels [15,16,17,18,19]. The favorable reaction conditions and high hydrolytic stability of acylhydrazine have proven particularly promising for the application of these hydrogels in the field of biomaterials [20,21,22,23]. However, the low reaction rate, and the locking of reversible reactions under neutral/weak base conditions severely limit the practical applications of these hydrogels [24,25]. Typically, arylhydrazone-based hydrogels are prepared over long periods depending on the reaction conditions, the self-healing of hydrogels also takes several hours [17,26]. Compared with the acylhydrazone bond, the boronic ester bond is more chemically active and can form with fast reaction kinetics [27,28,29]. Typically, boronate ester-based dynamic hydrogels can form in a few seconds and achieve self-healing on a short time scale. Moreover, the hydrogels crosslinked by boronate esters are multi-responsive to pH, sugar, and other diol-containing molecules. In fact, boronate ester-based materials have been constructed and used in drug delivery, saccharide sensing, memory materials, and injectable materials [30,31,32,33,34]. However, the low stability of the boronic ester bond prevent their practical applications [27,35]. It was demonstrated that the boronate ester bond and the arylhydrazone bond are compatible with each other. Incorporating these two dynamic bonds into a hydrogel will endow the hydrogel with some special properties.

Herein, we designed a new double DCB-crosslinked hydrogel based on the combination of the acylhydrazone and boronate ester bonds. The functional phenylboronic- and ketone-containing diblock copolymers P(*N*-isopropyl acrylamide-co-*N*-acryloyl-3-aminophenylboronic acid)-b-(*N*-isopropyl acrylamide-co-diacetone acrylamide) (PAD) with controllable composition and degree of polymerization (DP) were prepared via sequential two-step reversible addition−fragmentation chain transfer (RAFT) polymerization. The double DCBs-crosslinked hydrogels were constructed by the interaction of PAD with polyvinyl alcohol (PVA) and adipic dihydrazide (ADH) without any external stimuli. The synergistic advantages of integrating the active boronate ester and stable acylhydrazone bond in gelation process, self-healing ability, multi-responsive properties and the tunable mechanical property of the hydrogel were studied comprehensively.

## 2. Materials and Methods

### 2.1. Materials

*N*-isopropyl acrylamide (NIPAM, J&K Scientific, 99.5%, Beijing, China) was recrystallized three times in n-hexane and methylbenzene to remove inhibitor. Diacetone acrylamide (DAA, Macklin Co., 99%, Shanghai, China) was recrystallized from n-hexane for three times to remove inhibitor. 2,2’-Azobis (isobutylronitrile) (AIBN, J&K Scientific, 98%, Beijing, China) was recrystallized from ethanol. *N*-acryloyl-3-aminophenylboronic acid (AAPBA) and chain transfer agent *S*-1-Dodecyl-*S′*-(R,R′-dimethyl-R′′-acetic acid) trithiocarbonate (DDMAT) was synthesized and purified according to the literature procedures [36,37]. Acryloyl chloride, aminophenylboronic acid (APBA), adipic dihydrazide (ADH), poly(vinyl alcohol) (PVA, polymerization degree: 1750, hydrolysis degree: 87-89%) were purchased from J&K Scientific, Beijing, China. Sodium phosphate dibasic, sodium dihydrogen phosphate, methanol, dimethyl sulfoxide (DMSO), triethyl amine (TEA) and acetone were purchased from Sinopharm Chemical Reagent Co. Ltd., Beijing, China.

### 2.2. Synthesis of Diblock Copolymers

*N*-isopropyl acrylamide (NIPAM) (5.09 g, 45 mmol), *N*-acryloyl-3-aminophenylboronic acid (AAPBA) (0.96 g, 3.0 mmol), DDMAT (0.365 g, 1.0 mmol), AIBN (3.3 mg, 0.02 mmol) were dissolved in DMSO (15.6 mL) in a 25-mL reaction tube equipped with a magnetic stirrer. The mixture was degassed via three freeze-pump-thaw cycles, and then the tube was immersed in an oil bath at 70 °C. The polymerization was performed for 24 h and then terminated by rapid cooling through the immersion of the flask in liquid nitrogen. PAD2 was obtained after dialysis and lyophilization. Three macro-CTA with different boronic acid contents (6, 10, and 10% for PA1, PA2, and PA3, respectively) and segment lengths (50, 50, and 100 for PA1, PA2, and PA3, respectively) were synthesized in the same manner.

The obtained PA2 macro-CTA (2.78 g), diacetone acrylamide (DAA) (0.82 g, 2.5 mmol), NIPAM (2.55 g, 22.5 mmol), and AIBN (1.65 mg, 0.01 mmol) were dissolved in a mixed solvent of 95/5 DMSO/methanol solution (14 mL) in a 25-mL reaction tube flask equipped with a magnetic stirrer. After degassing by three freeze-vacuum-thaw cycles, the polymerization was performed at 70 °C for 36 h and then terminated by rapid cooling through the immersion of the flask in liquid nitrogen. The product was obtained after dialysis and lyophilization, and it was named as PAD2. Three functional diblock copolymers with different ketone contents (6, 10, and 10 for PAD1, PAD2, and PAD3, respectively) and second segment lengths (50, 50 and 100 for PAD1, PAD2, and PAD3, respectively) were prepared by following the procedure summarized in Appendix A.

### 2.3. Preparation of Hydrogels

The PAD@PVA@ADH hydrogels were constructed by homogeneously mixing PAD solutions, and the solution of polyvinyl alcohol (PVA) and adipic dihydrazide (ADH) at 4 °C, all the preparations were performed in the sodium phosphate dibasic/sodium dihydrogen phosphate aqueous buffer at pH 8.6. For example, a homogeneous 10% *w/v* PVA solution was prepared by dissolving PVA powder in aqueous buffer at 90 °C for 2 h. Next, ADH was added into the PVA solution (0.5 mL) to prepare the mixed solution. The mixed solution was transferred a PAD (20% w/v) solution (0.5 mL) while stirring, and the hydrogel was obtained. Other control groups used in this work were prepared in the same manner.

### 2.4. Characterization

Nuclear magnetic resonance (NMR) tests were conducted by using a Bruker Avance 400 M NMR (Billerica, MA, USA) at 298 K. Gel permeation chromatography (GPC) was performed using CBM-20A communication bus module (Kyoto, Japan) equipped with a RID-10A refractive index detector and LC-20AT liquid chromatograph from Shimadzu. Samples were passed through three series SB-802.5HQ, SB-806HQ and SB-804HQ columns using 3/1 methanol/water solution eluent at a flow rate of 0.5 mL min^−1^. The number-average molecular weight (*M_n_*) and polydispersity index (PDI) data are reported relative to polystyrene standards. Turbidity characterizations were conducted with a Beijing TU-1901 (Beijing, China) double beam UV-vis spectrophotometer in 0.5 mg mL^−1^ copolymer solutions at 500 nm wavelength, the transmittances were recorded after the temperature was held for 3 min. Rheological behavior of the hydrogels was characterized by a HAAKE MARS III (Waltham, WA, USA) rheometer, equipped with a cone–plate (C35/1Ti) geometries. Scanning electron microscopy (SEM) image was observed on a JSM-7500 (Kyoto, Japan) microscope to determine the morphology of the micro porous material with operating voltage at 10 kV. Mechanical tensile tests of the hydrogels were performed using a uniaxial mechanical testing device (Xi’an, China) at a stretch rate of 50 mm min^−1^.

## 3. Results and Discussion

### 3.1. Synthesis and Characterization of Diblock Copolymers

The diblock copolymers composed of a phenyboronic acid-bearing (PA) segment and a ketone-bearing (PD) segment were prepared via sequential two-step RAFT polymerization (Scheme 1). The unimodal peak of the GPC curve and narrow molecular weight distribution indicated that copolymerization proceeded in a controlled fashion (Appendix A). The chemical structure of the block copolymers was determined by comparing the corresponding peak areas of ^1^H NMR spectra as shown in Figure 1. Comparison of the integrated areas of the signals at 7.2–8.1 ppm (aromatic protons of PAAPBA) and 4.0 ppm (–N–CH– protons of PNIPAM) allowed the calculation of the number-average degree of polymerization (DP) for the macro-CTA of PA. The DP values of the diblock copolymers were calculated by comparing the signals for the PNIPAM protons, the signals at 2.2 ppm (–CH_3_ protons of the PDAA), and the signals at 1.3–1.9 ppm (–CH– protons of the backbone). The molecular weight of the polymers, as determined by GPC, was in good agreement with the theoretical number-average molecular weight calculated based on the monomer conversion [38]. Based on the above procedure, copolymers with different reactive group contents and segment lengths were synthesized, and the detailed information for these copolymers is summarized in Table 1.

As reactive copolymer, the thermo-, pH- and fructose-sensitivity of PAD were investigated comprehensively, as shown in Appendix A. To study the reaction of PAD with ADH and PVA, the mechanical property of single acylhydrazone- and boronate ester-crosslinked hydrogels were studied by oscillatory rheological experiments. The concentration of PAD2 was fixed to 10 wt%. Figure 2a shows the storage and loss modulus of single acylhydrazone-crosslinked hydrogels with different molar ratios of ketone groups (from PAD2) to hydrazine groups (from ADH). With the molar ratio of ketone to hydrazine varying from 4:1 to 1:1, the storage and loss modulus of the hydrogels increase from 574 and 21.4 Pa to 1121 and 43 Pa, reaching the maximum modulus. Continuing to increase the molar ratio of ketone to hydrazine leads to a significantly decrease in the modulus, indicating that the excess of the hydrazine groups results in the partial dissociation of the acylhydrazone-linkages. The rheological properties of single boronate ester-crosslinked hydrogels with different mass ratios of the PVA to PAD2 were also studied as shown in Figure 2b. The storage and loss modulus of the hydrogels increased drastically from 189 and 81 Pa to 2217 and 224 Pa with the mass ratio of PVA to PAD varying from 0.05 to 0.5, indicating that the dynamic reaction between the phenylboronic acid (from PAD2) and 1,3-diols (from PVA) is the driving force for hydrogel formation.

### 3.2. Gelation Process

The gelation processes of single arylhydrazone-based hydrogels formed from PAD and ADH were characterized using oscillatory time sweep experiments at different times during the gelation. The crossover time of the storage and loss moduli curves for the hydrogel formation reaction was used to characterize the gelation kinetics. Hydrogels prepared using PAD2 and ADH were selected as the model materials with the measurement results shown in Figure 3a. The crossover time decrease to 33,680, 15,760, and 8121 s as the solid content increased to 10, 15, and 20 wt%, respectively, indicating that polymer concentration plays a key role in gelation kinetics. The long gelation time is due to the fact that the hydrogel was prepared at pH 8.6, where the formation rate of the acylhydrazone bond is extremely low [17,26]. The critical gelation concentration of arylhydrazone-based hydrogels is above 5 wt%, 3.5 wt%, 3.5 wt%, for PAD1@ADH, PAD2@ADH, and PAD3@ADH, respectively. The crossover times of the hydrogels (4 × 3) with different PAD copolymers and various concentrations were carefully studied as shown in Appendix A, and it was found that higher reactive group content and higher segment length were related to fast gelation kinetics. Compared with the arylhydrazone-based hydrogels, the boronate ester-crosslinked hydrogel demonstrated a rapid gel kinetics. The gelation was occurred during the mixing of the PAD2 and PVA solutions, even before the sweeping started, resulting from the fast formation rate of boronate ester bond and the large molecular weight of PVA. According to our design, the ideal hydrogel networks are crosslinked by both the active boronate ester bonds and the slow acylhydrazone bonds, such that the PAD@PVA@ADH hydrogels demonstrate both properties. Benefiting from the fast reaction kinetics of the boronate ester bonds, the gelation occurred immediately upon homogeneous mixing of PAD with the PVA and ADH solutions as shown in Figure 3b. The storage modulus is reached 500 Pa in 300 s. The gelation process of the hydrogel has not yet been completed and the storage modulus has not reached the maximum value.

The time-dependent gelation processes of hydrogels were systematically studied by measuring the storage modulus at different time during gelation as shown in Figure 3c. The storage modulus of single boronate ester-crosslinked hydrogels increased rapidly and reached a plateau within two hours. The balanced value for storage modulus is about 2271 Pa, and it is much greater than the value at 300 s (Figure 2b). This is caused by the gradually increasing crosslink density in the process, while the storage modulus of single arylhydrazone-based hydrogels began to increase at 8 h and reached plateau for another 6 h. Interestingly, the process of increasing the storage modulus of the double DCBs-crosslinked hydrogel can be divided into two stages, corresponding to the fast formation of boronate ester bond and slow formation of acylhydrazone bond. In this case, the fast interaction enables the rapidly formation and maintain the gel state while the slow interaction can enhance the mechanical strength of the hydrogel. Moreover, the contribution of the two interactions to mechanical strength of the hydrogel can be accurately estimated. The double DCBs-crosslinked hydrogel showed a storage modulus of 2310 Pa on the first platform, which is similar to 2271 Pa of single boronate ester-crosslinked, while, on the second platform, storage modulus increased by 1507 Pa, which is higher than modulus of 1121 Pa for the single acylhydrazone-crosslinked hydrogel. The storage modulus during gelation of the PAD1@PVA@ADH and PAD3@PVA@ADH hydrogels were also studied as shown in Appendix A, and the results were similar to those of PAD2@PVA@ADH.

### 3.3. Mechanical Property

To investigate the mechanical properties of the hydrogels, oscillatory rheological measurements were performed at frequency sweep modes as shown in Figure 4a. Typical frequency-dependent curves of the modulus were observed for the single boronate ester-crosslinked hydrogels. In low angular frequency region, the hydrogels behaved liquid-like behavior that the loss modulus was greater than storage modulus, whereas, at high frequency, the hydrogel behaved like an elastomer with the storage modulus exceeding the loss modulus. Moreover, a crossover of the storage and loss modulus was observed, where the lifetime of the boronate ester bond was equal to the time scale probed in the experiment. The relaxation rate of the hydrogel was 0.024 s^−1^, obtained from the crossover frequency [39,40]. For acylhydrazone-crosslinked hydrogels, the storage modulus exceeded loss modulus throughout the studied frequency range, and both the storage and loss modulus behaved almost frequency-independent behavior. These results implied that the relaxation rate of the arylhydrazone bond crosslinker is extremely low. The double DCBs-crosslinked PAD2@PVA@ADH hydrogels demonstrated clearly reversible characteristics, with the frequency-dependent modulus curves observed, the storage modulus decreased and loss modulus increased at low frequency. Compared with the PAD2@PVA hydrogel, the lifetime of the double dynamic cross-links in the PAD2@PVA@ADH hydrogel is at a longer time scale. The storage modulus and loss modulus have a crossover tendency at the low frequency beyond the test range, indicating that the double DCBs-crosslinked hydrogel is stronger than boronate ester-crosslinked hydrogel under the same condition. The slow acylhydrazone linkage in hydrogels turned less dynamic in nature.

The concentration also has a significant effect on the mechanical strength. The increase in the polymer concentration led to an increase in the storage and loss moduli of the hydrogels as shown in Appendix A. For PAD2@PVA hydrogels, the crossover frequency of the storage and loss modulus curves were independent of the concentration and its value is approximately 0.024 Hz. This is likely attributable to the fact that polymer concentration did not alter the lifetime of the dynamic covalent crosslinkage [41]. However, for double DCBs-crosslinked hydrogel, with the increase of the solid content, the more rigid of the hydrogel was apparated due to the increase in the ratio of the slow acylhydrazone-linkage in hydrogel.

To gain better insight into the mechanical properties of the double DCBs-crosslinked hydrogel, the storage modulus of the hydrogels (3 × 6) with different compositions and various solid contents were tested and summarized in Figure 4b. The mass ratios of the PVA to PAD are fixed to 0.5 and the molar ratios of the ketone to hydrazine in hydrogels are fixed to 1. To evaluate the contribution of each cross-linkage to mechanical properties of hydrogels, the 2-h and 24-h storage modulus of the formation process were measured, corresponding to the contribution of the boronate ester formation and the whole storage modulus of the hydrogels. Moreover, the storage modulus of the single boronate ester- and single acylhydrazone-crosslinked hydrogels were also determined as shown in Appendix A. The complexes of the PAD and PVA contributed most of the mechanical strength of the hydrogel, especially at low solid content. With the increase of the solid content, the contribution of the acyhydrazone-linkages to mechanical strength increases rapidly, which is due to the rapid increase of stable acylhydrazone-linkages. However, the high viscosity and low mobility of the PVA solution restrict the strength of the hydrogels because of the low content of arylhydrazone even at the polymer concentration of 10 wt% (maximum concentration for obtaining homogeneous hydrogels in experiments) in the PAD@PVA@ADH hydrogels. For the enhanced modification of the mechanical properties, the mass ratios of the PVA to PAD of the hydrogels are kept constant at 0.1 (Figure 4c). The polymer concentration reached 15 wt%, and the storage modulus reached the relatively high values of 2.2, 4.9, and 5.3 kPa for PAD1@PVA@ADH, PAD2@PVA@ADH, and PAD3@PVA@ADH, respectively. Meanwhile, the arylhydrazone bond crosslinker in the hydrogels contributed to more than 80% of the storage modulus when the solid content exceeded 15%. Moreover, the longer segment length (comparing the PAD1@PVA@ADH and PAD2@PVA@ADH hydrogels) and higher contents of reactive moieties in the copolymer (comparing the PAD2@PVA@ADH and PAD3@PVA@ADH hydrogels) also promote mechanical strength.

We then further tested the stretch properties of the hydrogels. As shown in Figure 4d, the double DCBs-crosslinked hydrogel exhibits a higher breaking stress than the single DCB-crosslinked hydrogel. The tensile testing showed that the breaking stress of the gels were 33 kPa, 28 kPa and 55 kPa, for PAD2@PVA, PAD2@ADH, and PAD2@PVA@ADH, respectively.

### 3.4. Self-Healing Properties

The acylhydrazone bond-based hydrogel do not undergo self-healing due to the kinetically locked under weak base condition [15,42]. In this work, we incorporated active boronate ester and stable acylhydrazone bond in hydrogels, which would facilitate the hydrogel self-healing macroscopically. To qualitatively evaluate the self-healing ability, cut/heal tests were performed on hydrogels. As shown in Figure 5a, disks of the hydrogels were sliced into two halves and colored by carmine (red) or methylene blue (blue). The hydrogel heaves were placed together and contacted without external treatment. As expected, the PAD@ADH hydrogel did not self-heal after contacted for 24 h, due to the fact that the exchange reaction of the acylhydrazone bond was locked under weak base conditions. The boronate ester-crosslinked PAD@PVA hydrogel showed excellent self-healing ability with the cut line almost disappearing in 30 min because of the fast exchange reaction of the boronic ester bond. The double DCBs-based hydrogels PAD@PVA@ADH also demonstrated macroscopic self-healing, and the halves merged autonomously into a whole in 30 min.

To quantitatively evaluate the self-healing behavior of the hydrogel, the rheological recovery tests were performed. The hydrogel was subjected to increasing strain to induce a large-amplitude deformation (3000% strain) far beyond the linear viscoelastic region (Appendix A), followed by a recovery period under small-amplitude (1% strain) deformation conditions. To evaluate the self-healing ability, we calculated the healing efficiency (*HE*) of the hydrogel after each cycle. The healing efficiency refers to the ratio of storage modulus (Gh) of each cycle to the initial storage modulus (Gi) of the hydrogel. As shown in Figure 5c, the boronate ester-crosslinked hydrogel exhibited 100% healing efficiency after strain-induced disruption for three cycles, while the acylhydrazone-crosslinked hydrogel could only recover 39.6% under the same condition (Figure 5b). Interestingly, the double DCBs-crosslinked hydrogel recovered 85.6% storage modulus (Figure 5d), caused by the different exchange rate of the boronate ester and arylhydrazone. We believed that the hydrogel with more boronate ester-crosslinks contribute to rapid self-healing of the gel at the macro level and enhance the healing efficiency. Therefore, to improve the healing efficiency of double DCBs-crosslinked hydrogels, increased PVA content in hydrogel was adopted (Appendix A). After three damaging–recovery cycles, the healing efficiency was increased from 51.5% to 85.6% when the mass ratio of PVA to PAD2 increased from 0.1 to 0.5 (Appendix A).

### 3.5. Stimuli Responsiveness

As representative DCBs, both the acylhydrazone and boronic ester bond are reversible and highly pH-dependent. Thus, double DCBs-based hydrogels could undergo gel-sol-gel phase transition by regulating the pH (Figure 6a). As expected, the hydrogels dissolved into solution upon the addition of 20 μL of 5 M HCl in 1.0 mL hydrogels, and reverted to hydrogel state as quantitative amounts of N(C_2_H_5_)_3_ were added to the solution. Moreover, this transition was reversible and could be repeated for several cycles. However, the PAD2@PVA, PAD2@ADH, and PAD2@PVA@ADH hydrogels became unstable after seven, five, and seven cycles (Appendix A), respectively. This phenomenon may be attributed to the change in the volume of the hydrogels or to the effect of the salt.

Moreover, the acylhydrazone linkages can be dissociated by the addition of excess hydrazide-functional compounds [43]. As showed in Appendix A, after adding 8 equiv. ADH, the PAD2@ADH hydrogel was completely dissociated into a viscous liquid within 60 min, caused by the cleavage of the crosslinks. The boronate ester bond can be dissociated using another competitive diol-containing molecule [11,44]. The fructose was selected for tests owing to the higher affinity to complexing with PBA [27,35]. As showed in Appendix A, the PAD2@PVA hydrogel turned into a viscous liquid in 30 min with the addition of 50 equiv. fructose. One of the cross-linkages was failure by the chemical stimuli, and the double DCBs-crosslinked hydrogel could maintain the gel state. When the two chemical stimuli were applied, the double DCBs-crosslinked hydrogel converted into liquid as shown in Figure 6b. To further study the stimuli responsiveness, oscillatory rheological tests were performed (Figure 6c). The storage modulus of double DCBs-crosslinked hydrogel treated with excess of ADH was 2328 Pa, which is similar to contribution of the boronate ester linkages in hydrogel. Meanwhile, the storage modulus of fructose treated double DCBs-crosslinked hydrogel was also close to the contribution of acylhydrazone linkages in hydrogel. The results indicated that the combination of the acylhydrazone and boronate ester bond in hydrogel has enhanced the level of responsiveness.

### 3.6. Tunable Mechanical Property of the Hydrogel

The mechanical property of the double DCBs-crosslinked hydrogel can be readily engineered by selectively controlling one of the dynamic linkages. One of the dynamic linkages maintains the gel state, while the other is partial formation or dissociated by the control, thus achieving the tunable mechanical property of the hydrogel. To this end, we assessed the mechanical property of the hydrogels with various molar ratios of ketone to hydrazine. The PAD2 concentration is fixed to 10 wt% and the mass ratio of the PVA to PAD2 of the hydrogel is fixed to 0.5. The storage modulus of the PAD2@PVA@ADH hydrogels with various molar ratios of ketone to hydrazine are summarized in Figure 7a. It was observed that the maximum storage moduli of 1358 Pa appeared at the [ketone]/[hydrazine] molar ratio of 1, with the [ketone]/[hydrazine] molar ratio exceeding 1; the storage modulus decreased, which may be attributed to the acylhydrazone exchange reaction caused by excess ADH. The dissociation of partial acylhydrazone bond results in the decrease of crosslinking density. With the [ketone]/[hydrazine] molar ratio reached 8.0, the storage modulus decreased to 371 Pa, which is almost equal to the singly boronate ester-crosslinked hydrogel. These results indicated that the mechanical strength could be tuned over a large range. The interior morphologies of the hydrogels with various [ketone]/[hydrazine] molar ratios are shown in Appendix A. Smaller pore size and thicker pore walls were observed with the molar ratios of ketone to hydrazine close to 1.

The mechanical strength of the hydrogel can also be tuned by partially dissociating the boronate ester linkages. The PAD2 content is fixed to 10 wt% and the molar ratio of hydrazine to ketone is fixed to 1.0. The storage modulus decreased with the increase of the addition of the fructose. With the [phenylboronic acid]/[fructose] molar ratio exceeding 16, the storage modulus of the hydrogels has been basically unchanged. However, the storage modulus of the hydrogels of the doubly DCBs-crosslinked hydrogel treated with the excess of the fructose is slightly higher than that of singly acylhydrazone-crosslinked hydrogel, which may be attributed to the high molecular weight of PVA molecules in hydrogel.

## 4. Conclusions

In summary, we described the design of doubly dynamic hydrogels containing both acylhydrazone and boronate ester bonds. The active boronate ester linkage endows the hydrogel with fast gelation kinetics and self-healing ability, and the stable acylhydrazone linkage can enhance the mechanical property of the hydrogel. Compared with its singly crosslinked hydrogels, the level of stimuli-responsiveness of the hydrogel has been enhanced. Moreover, the combination of the dual dynamic covalent bonds in hydrogel enhances the regulation and control level of the mechanical properties. The mechanical property of the hydrogel can be readily engineered not only by changing the network architecture and composition, but also by alternately controlling the formation or dissociation of one of the dynamic linkages. Based on these results, we believe that this work provides a promising strategy for designing smart hydrogels with multiple responsiveness and tunable properties.

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
