# Peer review of "Doubly Dynamic Hydrogel Formed by Combining Boronate Ester and Acylhydrazone Bonds"

_polymers, 2020, doi:10.3390/polym12020487_

Round 1
Reviewer 1 Report
The article demonstrate a double network made of two crosslinking compounds with very different kinetics of formation.
For the characterization the authors rely on the gelation, and more precisely on the elastic shear modulus.
As non-specialist of polymer synthesis, it is somewhat difficult to judge the novelty of the double network. However, I believe this to be an interesting paper, fit for publication in Polymers.
Some modification and clarification of the manuscript will be welcomed. Follows a number of comments that could be used to modify the manuscript.
- In particular I suggest the authors try to find a way to simplify the acronyms of the compounds to make the understanding of the double network composition simpler. As is, I found it very difficult to follow, especially to identify the difference between the samples.
- Overall it was not so clear to me what are the benefits brought by the complex formulation. Does the new properties justify the extra complexity ?
- On the double dynamic name : Is this the most appropriate. I rather understand double dynamics with two components having different dynamics, like glassy and rubbery . Here it seems we are more in presence of two different kinetics , with one gel forming faster than the other one, but not necessarily with distinct equilibrium dynamics feature.
- There is no description on how the tensile test were made (fig4d)
- On the healing properties: It is not that clear to me what the reported healing efficiency tests measures and if it relates to healing ? how sure can one be that the gel fractured during the large strain shear ? how do you measure that ? could it be something else like slip ? fig5 does not allow to understand how much the gel was affected by the large strain . Showing the strain on a linear scale might help. It seems to me that the value of G' and G'' go back very fast to the pre-strain ones. Could it be that the gel was not really broken ?
- on the morphology discussion: the authors mentioned fig7b that does not exist. They also discussed morphology that is only shown in SI fig13.
Author Response
Response to Reviewer 1 Comments
Comment
The article demonstrate a double network made of two crosslinking compounds with very different kinetics of formation.
For the characterization the authors rely on the gelation, and more precisely on the elastic shear modulus.
As non-specialist of polymer synthesis, it is somewhat difficult to judge the novelty of the double network. However, I believe this to be an interesting paper, fit for publication in Polymers.
Some modification and clarification of the manuscript will be welcomed. Follows a number of comments that could be used to modify the manuscript.
Response:
We really appreciate your constructive comments. Thank you again for all your efforts on our manuscript. In accordance with your valuable suggestions, we have revised our manuscript (see details below).
- In particular I suggest the authors try to find a way to simplify the acronyms of the compounds to make the understanding of the double network composition simpler. As is, I found it very difficult to follow, especially to identify the difference between the samples.
Response 1: Thank you very much for your constructive suggestion. We are so sorry for the ambiguous acronyms of the compounds. In order to make readers better understand the double network composition and identify the difference between the samples. We checked our manuscript carefully and corrected all typos and ambiguous acronyms. Moreover, we put the characterization of copolymers in Table S1 to the text.
- Overall it was not so clear to me what are the benefits brought by the complex formulation. Does the new properties justify the extra complexity ?
Response 2: We really appreciate your kind comments. In this study, the hydrogels were prepared by combining two kinetically distinct dynamic covalent bonds, boronate ester and acylhydrazone bonds. The integration of dual or multiple DCBs in one system can enhance the performance of dynamic hydrogels.
The difference in kinetics endows that the contribution of each linkage to mechanical strength of the hydrogel can be accurately estimated. In this case, the fast interaction enables the rapidly formation and maintain the gel state while the slow interaction can enhance the mechanical strength of the hydrogel. Compared with the single DCB-based hydrogels, the storage modulus of the double DCB-based hydrogels is higher than the sum of the two single DCB-based hydrogels. For example, the storage modulus of the PAD2@ADH@PVA, PAD2@ADH, PAD2@ PVA are 3817 Pa, 2271 Pa and 1121 Pa respectively.
The hydrogel inherits the stimuli responsiveness of boronate ester and acylhydrazone linkages, and the two linkages can be destructed independently. Compared with the single DCB-based hydrogels, the level of responsiveness of in hydrogel has enhanced. The mechanical property of the hydrogel can be readily engineered by controlling one of the dynamic linkages form or dissociate.
- On the double dynamic name: Is this the most appropriate. I rather understand double dynamics with two components having different dynamics, like glassy and rubbery. Here it seems we are more in presence of two different kinetics, with one gel forming faster than the other one, but not necessarily with distinct equilibrium dynamics feature.
Response 3: Thank you for your kind comments. In our manuscript, dynamic hydrogel means this material is fabricated by dynamic covalent bonds (DCBs). Dynamic covalent bond is different from common covalent bond, and it can selectively undergo reversible breaking and reformation without irreversible side reaction. Both of the boronate ester and acylhydrazone bonds are representative DCBs, although kinetics of two dynamic covalent bonds are different. Indeed, both the boronate ester and acylhydrazone bonds have been widely used to constructed doubly dynamic hydrogels. For example, Wei, Z.; Yang, J.H.; Liu, Z.Q.; Xu, F.; Zhou, J.X.; Zrínyi, M.; Osada, Y.; Chen, Y.M. Novel Biocompatible Polysaccharide‐Based Self‐Healing Hydrogel. Advanced Functional Materials 2015, 25, 1352-1359, doi:10.1002/adfm.201401502, and Collins, J.; Nadgorny, M.; Xiao, Z.; Connal, L.A. Doubly Dynamic Self-Healing Materials Based on Oxime Click Chemistry and Boronic Acids. Macromolecular Rapid Communications 2017, 38, 1600760, doi:10.1002/marc.201600760.
- There is no description on how the tensile test were made (fig4d)
Response 4: Thank you for your kind comments. We are very sorry for making this mistake. Mechanical tensile tests of the hydrogels were performed using a uniaxial mechanical testing device at a stretch rate of 50 mm min-1. The modifications were marked red color (in line 147-149).
- On the healing properties: It is not that clear to me what the reported healing efficiency tests measures and if it relates to healing ? how sure can one be that the gel fractured during the large strain shear ? how do you measure that ? could it be something else like slip ? fig5 does not allow to understand how much the gel was affected by the large strain . Showing the strain on a linear scale might help. It seems to me that the value of G' and G'' go back very fast to the pre-strain ones. Could it be that the gel was not really broken ?
Response 5: Thank you very much for your constructive suggestion. In this study, the rheological recovery tests were performed to test the self-healing behaviour of the hydrogels. The healing efficiency was used to evaluate the self-healing ability of the hydrogels. The hydrogel was subjected to increasing strain to a large amplitude deformation (3000% oscillatory strain) far beyond the linear viscoelastic region, followed by a recovery period under small-amplitude (1% oscillatory strain) deformation conditions. The storage modulus dominates the linear viscoelastic region and the loss modulus dominates the large strain range far beyond the linear viscoelastic region. At large strain shear, the hydrogels behaved liquid-like behavior that the loss modulus was greater than storage modulus, indicating that the hydrogel network has been destructed. The healing efficiency refers to the ratio of storage modulus (Gh) of each cycle to the initial storage modulus (Gi) of the hydrogel. In the experiment, there was no slip and other factors affecting the self-healing was observed.
To demonstrate the influences of the large strain on the hydrogels, the oscillation amplitude sweeps were performed. In linear viscoelastic region, the storage modulus remained above loss modulus, indicating that the hydrogel networks were rigid and that the networks remained intact in this strain amplitude. With increasing strain amplitude, gel-liquid transition points, where the loss modulus curve crosses over the storage modulus curve occurred, indicating the beginning of the destruction of the hydrogel network. Indeed, this approach of the damaging–recovery to has been widely used to evaluate the self-healing ability of the hydrogel, for example: Jay, J.I.; Langheinrich, K.; Hanson, M.C.; Mahalingam, A.; Kiser, P.F. Unequal stoichiometry between crosslinking moieties affects the properties of transient networks formed by dynamic covalent crosslinks. Soft Matter 2011, 7, 5826-5835, doi:10.1039/C1SM05209. Deng, C.C.; Brooks, W.L.A.; Abboud, K.A.; Sumerlin, B.S. Boronic Acid-Based Hydrogels Undergo Self-Healing at Neutral and Acidic pH. ACS Macro Letters 2015, 4, 220-224, doi:10.1021/acsmacrolett.5b00018. Vatankhah-Varnoosfaderani, M.; Ghavaminejad, A.; Hashmi, S.; Stadler, F.J. Mussel-inspired pH-triggered reversible foamed multi-responsive gel--the surprising effect of water. Chem Commun (Camb) 2013, 49, 4685-4687, doi:10.1039/c3cc41332b.
Figure S11. Oscillatory amplitude sweeps at 1 Hz
The storage modulus and loss modulus go back very fast to the pre-strain ones. the boronate ester-crosslinked hydrogel exhibited 100% healing efficiency after strain-induced disruption for three cycles, while the acylhydrazone-crosslinked hydrogel could only recover 39.6% under the same condition, indicating that the different exchange rate of the boronate ester and acylhydrazone bonds. Moreover, the hydrogels behaved liquid-like behavior that the loss modulus was greater than storage modulus, indicating that the hydrogel network has been destructed.
- on the morphology discussion: the authors mentioned fig7b that does not exist. They also discussed morphology that is only shown in SI fig13
Response 6: Thank you for your suggestion. We are very sorry for making this mistake. We have amended this mistake. The interior morphologies of the hydrogels with various [ketone]/[hydrazine] molar ratios are shown in Figure S14. The modifications were marked red color (in line 367)

Reviewer 2 Report
The paper presents a comprehensive work on doubly dynamic covalent bounds. Although the gel formation is already well described in the literature, the paper presents an approach in combining two different (known) mechanisms to achieve higher gel strength/stiffness. The reported effect of (fast) self-healing was rather contributed by the boronate ester crosslinked structure.
Moduli in Figure 3: please explain the difference:
2b) storage modulus of PDA2@PVA (single boronate ester) and PDA2@PVA@AGH (double DCB) reached a level of ca 500 (bellow 1000) Pa after 300min = 6h 2c) storage modulus PDA2@PVA (single boronate ester) and PDA2@PVA@AGH (double DCB) reached a level of ca 2300 (higher than 2000) Pa after 2 hrs (also explained in text, linee 224-225
PDA2@PVA@AGH demonstrated clearly reversible characteristic…(lines 257-258)?: please explain further
Moduli in Figure 4: Fig. 4c (PVA:PAD ratio 0,1) shoes tendency higher storage moduli than Fig. 4b (PVA:PAD ratio 0,5), according to the authors, due to the low mobility of PVA in the solution (lines 281-282) It stands contradictory o findings in Fig 2b: please explain
Lines 298,299: kinetically locked ? please explain
Table S2: how can the healing efficiency greater than 100% ?
Line336-339: any mechanism for trimethylamine and how much was it added ?
Figure 6.: 6b: PAD@PVA@ADH + fructose + hydrazine: please add the data of this series to 6c
Please also run typo check, uncompleted sentences.
Author Response
Response to Reviewer 2 Comments
Comment
The paper presents a comprehensive work on doubly dynamic covalent bounds. Although the gel formation is already well described in the literature, the paper presents an approach in combining two different (known) mechanisms to achieve higher gel strength/stiffness. The reported effect of (fast) self-healing was rather contributed by the boronate ester crosslinked structure.
Response:
We really appreciate your constructive comments. Thank you again for all your efforts on our manuscript. In accordance with your valuable suggestions, we have revised our manuscript (see details below).
Moduli in Figure 3: please explain the difference:
2b) storage modulus of PDA2@PVA (single boronate ester) and PDA2@PVA@ADH (double DCB) reached a level of ca 500 (bellow 1000) Pa after 300min = 6h 2c) storage modulus PDA2@PVA (single boronate ester) and PDA2@PVA@AGH (double DCB) reached a level of ca 2300 (higher than 2000) Pa after 2 hrs (also explained in text, linee 224-225.
Response 1: We really appreciate your constructive comments. Figure 3b demonstrates the oscillatory time sweep experiments of the PDA2@PVA (single boronate ester) and PDA2@PVA@ADH (double DCB). The polymer solutions were mixed on the plate of the rheometer, and the experiments were performed at a frequency of 1.0 Hz and a strain of 1.0%. The gelation process of the hydrogel has not yet been completed and the storage modulus has not reached the maximum value in 300 s (5 min). As a result, the storage modulus is just 500 Pa at this moment. Figure 3c demonstrates the storage modulus of the hydrogels at different time during the gelation. The gelation process of the hydrogel almost completes and the storage modulus of the hydrogels increases rapidly and reaches a plateau within two hours. The balanced value for storage modulus is about 2300 Pa, and it is much greater than the value at 300 s. This is caused by the gradually increasing crosslink density in the process. The modifications were marked red color (in line 226-227 and line 226-227).
PDA2@PVA@AGH demonstrated clearly reversible characteristic…(lines 257-258)?: please explain further.
Response 2: Thank you very much for your kind advice. The double DCBs-crosslinked PAD2@PVA@ADH hydrogels demonstrated clearly reversible characteristic, with the frequency-dependent modulus curves were observed, the storage modulus decreased and loss modulus increased at low frequency. The modifications were marked red color (in line 257-258).
Moduli in Figure 4: Fig. 4c (PVA:PAD ratio 0,1) shoes tendency higher storage moduli than Fig. 4b (PVA:PAD ratio 0,5), according to the authors, due to the low mobility of PVA in the solution (lines 281-282) It stands contradictory o findings in Fig 2b: please explain
Response 3: Thank you so much for your kind comments. For Figure 4b (the mass ratios of PVA to PAD are fixed to 0.5), The complexes of the PAD and PVA contributed most of the mechanical strength of the hydrogel (bottom). However, the high viscosity and low mobility of the PVA solution restrict the increase of the solid contents of the hydrogels, the maximum concentration for obtaining homogeneous hydrogels was 10 wt% in experiments. For Figure 4c (the mass ratios of PVA to PAD are fixed to 0.1), the arylhydrazone bond crosslinker in the hydrogels contributed most of the mechanical strength of the hydrogel and the polymer concentration reached 15 wt%. At the same solid contents, the storage modulus of the hydrogels in Figure 4b is greater than hydrogels in Figure 4c. With the polymer concentration exceeded 10 wt% in Figure 4c, the storage modulus of the hydrogels reached the relatively high values, which caused by the rapid increase of stable acylhydrazone-linkages.
Lines 298,299: kinetically locked ? please explain
Response 4: Thank you so much for your kind comments. In weak base conditions, the dynamic exchange for acylhydrazone reaction slows down significantly.
Table S2: how can the healing efficiency greater than 100% ?
Response 5: Thank you so much for your kind comments. This may be due to the fast gelation of the boronate ester-based hydrogels, resulting in the inhomogeneous of the hydrogels for the first gelation process. The hydrogels become more homogeneous after repeated dynamic strain sweep tests.
Line336-339: any mechanism for trimethylamine and how much was it added ?
Response 6: Thank you so much for your kind comments. The N(C2H5)3 was added to neutralize the HCl. In every cycle, 20 μL of 5 M N(C2H5)3 was used, which was equal to the addition of HCl.
Figure 6.: 6b: PAD@PVA@ADH + fructose + hydrazine: please add the data of this series to 6c
Response 7: Thank you very much for your valuable suggestion. In Figure 6b, the ‘hydrazine’ means the hydrazide-functional compounds. In fact, 8 equiv. ADH was added to the hydrogels as shown in Figure 6b. The data of this series has been included in Figure 6c. We are so sorry for this ambiguous expression. We have improved Figure 6b to avoid misleading, and the modifications were marked red color (in line 367).
Figure 6. Photographs of phase transitions of PAD2@PVA@ADH hydrogel, (a) pH-triggered gel-sol-gel and (b) fructose and ADH. (c) Storage/loss modulus of the hydrogel under different chemical stimuli.
Please also run typo check, uncompleted sentences.
Response 8: Thank you for your suggestion. We are very sorry for making these mistakes. We checked our manuscript carefully and corrected all typos and uncompleted sentences.
